# Neuronal Hyperexcitability and Free Radical Toxicity in Amyotrophic Lateral Sclerosis: Established and Future Targets

**DOI:** 10.3390/ph15040433

**Published:** 2022-03-31

**Authors:** Kazumoto Shibuya, Ryo Otani, Yo-ichi Suzuki, Satoshi Kuwabara, Matthew C. Kiernan

**Affiliations:** 1Department of Neurology, Graduate School of Medicine, Chiba University, Chiba 260-8677, Japan; kazumoto@net.email.ne.jp (K.S.); naschan127@yahoo.co.jp (R.O.); yo.suzuki0531@gmail.com (Y.-i.S.); kuwabara-s@faculty.chiba-u.jp (S.K.); 2Brain and Mind Centre, Department of Neurology, University of Sydney, Royal Prince Alfred Hospital, Sydney 2050, Australia

**Keywords:** excitotoxicity, free radicals, cortical excitability, nerve excitability, amyotrophic lateral sclerosis

## Abstract

Amyotrophic lateral sclerosis (ALS) is a devastating disease with evidence of degeneration involving upper and lower motor neuron compartments of the nervous system. Presently, two drugs, riluzole and edaravone, have been established as being useful in slowing disease progression in ALS. Riluzole possesses anti-glutamatergic properties, while edaravone eliminates free radicals (FRs). Glutamate is the excitatory neurotransmitter in the brain and spinal cord and binds to several inotropic receptors. Excessive activation of these receptors generates FRs, inducing neurodegeneration via damage to intracellular organelles and upregulation of proinflammatory mediators. FRs bind to intracellular structures, leading to cellular impairment that contributes to neurodegeneration. As such, excitotoxicity and FR toxicities have been considered as key pathophysiological mechanisms that contribute to the cascade of degeneration that envelopes neurons in ALS. Recent advanced technologies, including neurophysiological, imaging, pathological and biochemical techniques, have concurrently identified evidence of increased excitability in ALS. This review focuses on the relationship between FRs and excitotoxicity in motor neuronal degeneration in ALS and introduces concepts linked to increased excitability across both compartments of the human nervous system. Within this cellular framework, future strategies to promote therapeutic development in ALS, from the perspective of neuronal excitability and function, will be critically appraised.

## 1. Introduction

Amyotrophic lateral sclerosis is a complex disease characterized by degeneration of the upper and lower motor neuron of the human nervous system. A recent national ALS registry study in the United States, with data collected between 2014 and 2016, identified age-adjusted incidence rates of 1.5 to 1.7 per 100,000 U.S. population, with ALS more common among whites, males and persons aged 60–79 years [1]. The median survival of ALS from the onset to respiratory failure is 2–5 years [2]. While Jean-Martin Charcot originally described ALS in the 1860s, the definitive processes that lead to disease onset are yet to be identified [3]. Patients with ALS frequently experience muscle cramps, fasciculations, weakness, progressive gait disturbance, dysphagia, weight loss, dysarthria and respiratory failure, while a significant proportion also develops cognitive difficulties and frontotemporal dementia [4].

While the clinical phenotype is well known, the underlying pathophysiology in ALS remains to be elucidated. While 5–10% of ALS patients have family history of ALS, the remaining 95% patients are classified as sporadic [5]. Multiple predisposing factors have been implicated in sporadic ALS, including genetic, environmental, traumatic, dietary, metabolic and occupational factors, although the interplay of these factors remains unclear [6]. At a molecular level, several pathophysiological changes have been reported, including glutamate-induced excitotoxicity, increased oxidative stress, neurofilament accumulation, ion channel and pump dysfunction, intracellular aggregation, such as SOD1 and TDP-43 protein, dysregulated RNA metabolism, mitochondrial dysfunction, disruption of axonal transport systems, immunological changes and proteasomal dysfunction [7,8]. Separate to disruption of the motor neuron, the supporting network of astrocytes and microglia may also contribute to neurodegeneration, linked to insufficient release of neurotrophic factors, secretion of neurotoxic mediators and modulation of glutamate receptor expression [9]. Combinations of these pathophysiological alterations have been linked to contribute to ALS pathogenesis [10,11,12,13,14].

In terms of therapeutic progress, a recent systematic review investigated Phase II, Phase II/III and Phase III clinical trials in ALS using trial registries and PubMed from 2008 to 2019, and it identified 125 trials that investigated 76 drug formulations with recruitment of more than 15,000 ALS patients [15]. Despite this large body of clinical trial activity, efficacious drugs that are available for ALS in clinical practice remain limited. Presently, two drugs, riluzole and edaravone, have been established as useful to slow disease progression in ALS [16]. Riluzole exerts cellular and anti-glutamatergic properties, with several randomized trials establishing the efficacy of riluzole in large populations of ALS patients, with typically a 3–6 month extension of survival [17]. Edaravone eliminates free radicals (FRs), and one randomized trial identified slowing of ALS functional deterioration over a 6 month period, with effects particularly in younger patients with minimal disease involvement [18]. Additionally, a recent non-randomized study also suggested that edaravone prolongs survival of ALS patients [19]. Based on these pieces of combined evidence, the role of excitotoxicity and exposure to FRs have been pursued as key pathophysiologies linked to the development of ALS, which will form the key theme of the present review.

## 2. Oxidative Stress in Motor Neuron Degeneration of ALS

Glutamate is the excitatory neurotransmitter, active in the brain and spinal cord, that binds to several inotropic receptors, including α-amino-3-hydroxy-5-methyl-4-isoxazolepropionate (AMPA) and *N*-methyl-D-aspartate (NMDA) receptors [20]. Excessive activation of these receptors, with defects in the clearance of these neurotransmitters from the synaptic cleft, combined with increased permeability of Ca^2+^, accumulates excitatory mediators, generates FRs and impairs neurons [21] (Figure 1). Neurotoxicity mediated by these excitatory mediators (termed excitotoxicity) may secondarily induce degeneration through the influx of calcium ions (Ca^2+^) [22]. This influx of Ca^2+^ into cells activates enzymes such as endonucleases, phospholipases and proteases, which cause neuronal injury and ultimately cell death [21].

It is well known that oxygen plays a pivotal role for all living organs, including neuronal cells, although the excessive presence of oxygen becomes harmful to various intercellular organelles [23]. As such, the usage and uptake of oxygen are strictly maintained and monitored via complex feedback systems within the cell. Separately, oxidative stress is produced by the imbalance between oxidants and antioxidants within this biological system. The excessive level of reactive oxygen species (ROS) or functional impairments of antioxidant regulatory systems result in this imbalance [24]. The univalent metabolic reduction status of oxygen has adverse effects including the generation of ROS, not only oxygen radicals but also non-radicals that are easily converted into radicals. FRs typically exhibit one or more unpaired electrons and may exist independently [25,26]. ROS may contribute to the degeneration of neuronal cells by modulating the function of target biomolecules, such as DNA, RNA, lipids, proteins and related processes, such as nucleic acid oxidation and lipid peroxidation in the cell. In turn, ROS released from damaged neurons influence surrounding glial cells [27]. Moreover, ROS disrupts glutamate transport in surrounding astrocytes, increases glutamate concentrations in the synaptic cleft, raises Ca^2+^ influx into the cells and mitochondria and leads to impaired glial function [27]. Additionally, ROS may activate glia and induce proinflammatory cytokine production. Eventually, these neurodegenerative processes spread from a focal lesion to involve a more diffuse neuronal population. Given that the brain itself consumes substantial oxygen, it substantially contributes to the production of FRs, with the consequent risk of triggering neurodegenerative processes.

Various kinds of ROS, including hydrogen peroxide, superoxide anions and hydroxyl radicals, are known to exert deleterious effects on both the central and peripheral nervous system. As an example, neurofilaments contribute essential cytoskeleton for the central nervous and peripheral nervous systems, with abnormal accumulation of neurofilaments a frequent accompaniment of ALS [28]. Linked to this accumulation, ROS produced by mice with SOD1-mutations have been reported mainly to bind to neurofilaments, disrupting the neurofilament proteins, while promoting neurofilament aggregation. ROS may contribute to neurodegeneration in ALS via such a process.

While considering oxidative stress and neurodegeneration in ALS [29], motor neurons are remarkably sensitive to this process for several reasons. First, motor neurons are large cells, approximately 100 microns in diameter, with axons up to 1 m in length. As such, metabolic demands are significant, with high oxygen demands and metabolism needs that result in increased production of ROS [30]. Secondly, motor neurons express low levels of calcium-binding proteins. As such, excessive calcium ions that enter mitochondria result in increasing ROS production. Thirdly, the expression of protective antioxidant proteins, such as catalase, is lower in the motor neuron than in other cell types [31]. The combination of these factors likely contributes to the processes of degeneration that involve the motor neuron in ALS [32].

Markers for protein oxidation and DNA oxidative damage are increased in the motor cortex of patients diagnosed with sporadic ALS. In further support, immunohistochemical studies established increased neuronal staining for enzymes inducible by oxidative stress, lipid peroxidation and DNA oxidative damage in the spinal cord of sporadic ALS and familial ALS [32]. Additionally, 4-hydroxy-2,3-nonenal (HNE), a product of membrane lipid peroxidation, was also increased in motor neurons, astrocytes and microglia cells of ALS patients [33]. Furthermore, these HNE levels are significantly elevated in the sera and cerebrospinal fluid of sporadic ALS patients [34], suggesting that increased oxidative stress significantly contributes to motor neuron degeneration in ALS.

In conclusion, excessive activation of glutamate receptors induces oxidative stress, and oxidative stress can cause neurodegeneration via various mechanisms. Moreover, oxidative stress increases glutamate concentrations and raises Ca^2+^ influx. As such, the excitatory neurotransmitter and oxidative stress are tightly connected with each other and might be substantially related with neurodegeneration in ALS.

## 3. Increased Cortical Excitability

### 3.1. Neurophysiological Evidence

Several neurophysiological techniques, especially transcranial magnetic stimulation (TMS), have identified features consistent with motor cortical hyperexcitability in ALS. In the 1980s, TMS was developed to induce electric currents in the brain through an insulated wire coil placed on the skull to evaluate human brain function [35]. This technique can non-invasively generate direct (D) and indirect (I) waves, which reflect brain network function within the primary motor cortex [36,37,38], using single and multiple-pulse TMS protocols [39,40,41].

Motor threshold (MT), which is typically measured by single pulse TMS, is usually defined as the stimulus which induces motor evoked potential (MEP) amplitudes. Resting MT is assumed to be influenced by sodium channel function, excitability of glutamatergic synapses and the number and density of cortico-cortical axons and corticospinal neurons [42,43,44,45]. Several studies have identified reduced MT in ALS, especially during early disease stages [46]. Reduced MT is consistent with cortical hyperexcitability and can be observed in ALS patients with hyper-reflexia, preserved muscle bulk and profuse fasciculations [47]. Fasciculations, characteristic features of ALS, are frequently seen in ALS patients, especially during the early stages, and suggest hyperexcitability of motor neurons [48,49,50,51]. Similarly, increased amplitudes of motor evoked potential (MEP) have been reported in ALS, most prominently in the early stage [52,53]. MEP amplitudes reflect the density of corticomotoneuronal projections onto motor neurons and are mediated by glutamatergic, noradrenergic and GABAergic neurotransmission. As such, increased MEP amplitudes are consistent with an increased excitability via alterations in these neurotransmitters. Supplementally, several lines of evidence have disclosed increased glutamatergic and noradrenergic and decreased GABAergic neurotransmitters in ALS patients [54,55,56]. These alterations, suggesting increased excitability, might contribute to increased MEP amplitudes.

The cortical silent period (CSP), also measured by single pulse TMS, is determined by cortical and spinal mechanisms [57,58], particularly GABA_B_ inhibitory neurons. Absence or shortened CSP duration has typically been reported in ALS, especially during the early phase of the disease, and are rarely observed in mimic neuromuscular diseases, including multifocal motor neuropathy (MMN), spinal muscular atrophy (SMA), Kennedy’s disease, post-poliomyelitis syndrome and distal hereditary motor neuronopathy with pyramidal features [59,60,61,62,63]. The reduction of CSP is potentially related with dysfunction of GABA_B_ inhibitory neurons.

Short interval intracortical inhibition (SICI), measured by paired pulse TMS, is a biomarker of inhibitory GABAergic intracortical circuits located within the primary motor cortex [64,65]. Numerous studies have established the presence of decreased SICI in sporadic and familial ALS cohorts, suggesting dysfunction of GABA_A_ inhibitory neurons within the motor cortex [66,67,68]. A complementary study investigated SICI in pre-symptomatic SOD-1 mutation carriers and found that SICI becomes reduced 6–9 months prior to symptom onset [53]. Additionally, this decreased SICI evolves with disease progression and spreads from clinically affected regions to involve previously spared regions [67,69]. Decreased SICI is a feature found in ALS and not in ALS mimic disorders, such as Kennedy’s disease, MMN, chronic inflammatory demyelinating polyneuropathy and SMA [70]. As such, decreased SICI is assumed to be a promising diagnostic biomarker that differentiates ALS from ALS-mimic disorders with high sensitivity and specificity [70].

Other features that support the advent of cortical hyperexcitability include intracortical facilitation (ICF), which is mediated by I waves, which originate at the cortical level through synaptic input from excitatory interneuronal circuitries onto corticomotoneuronal cells [64]. Increased ICF, suggesting increased intracortical excitability, is a typical finding in ALS [71], while somatosensory evoked potentials (SEP) also exhibited increased amplitudes [72,73], with sensory hyperexcitability associated with shorter survival [72].

### 3.2. Imaging Evidence

Spectroscopy provides information about the neurochemical composition of the brain and has been applied to ALS [74]. Results have identified altered glutamate/GABA neurotransmitter flux balance in ALS. When normalized for neuronal volume, several studies revealed elevated neuronal glutamate, suggesting increased activity of glutamatergic neurons [75,76]. This increase in neuronal glutamate concentration, combined with decreased neuronal volume, is assumed to be specific to ALS. Additionally, when normalized to glutamate levels, a decrease in neuronal GABA, suggesting a loss of inhibitory regulation, has been reported in several studies [77,78]. Moreover, the combination of increased neuronal glutamate and decreased neuronal GABA appears highly specific to ALS [74].

Radioligand neuroimaging studies using the GABA_A_ receptor ligand [^11^C]-Flumazenil have been reported in ALS [55,79,80,81], with widespread reductions in binding in sporadic and SOD1 D90A ALS patients, particularly involving the frontal lobes and anterior cingulate gyri [55,79,80,81]. The PET ligand [^62^Cu]-ATSM is a copper-linked small molecule known to distribute to areas of hypoxia and oxidative stress [82]. A PET study using this [^62^Cu]-ATSM showed increased tracer accumulation in the motor cortex, paracentral lobules and right superior parietal lobule in ALS compared to controls.

### 3.3. Pathological and Animal Model Evidence

In terms of animal models, a microdialysis study in the wild-type rat lumbar spinal cord revealed glutamate excitotoxicity [83]. Specifically, perfusion of AMPA elicited intense muscular spasms, followed by permanent hindlimb paralysis with a remarkable motor neuron degeneration, with effects prevented by co-perfusion with an AMPA receptor antagonist. Findings from this study suggest a susceptibility of motor neurons to Ca^2+^ dysregulation [84]. In ALS patients, GluA2 transcriptional editing of motor neurons becomes impaired, leading to increased Ca^2+^ permeability [85].

From a physiological perspective, glutamate is released into the synaptic cleft by the presynaptic neuron and non-neuronal cells such as astrocytes, with the extracellular concentration of glutamate regulated by rapid uptake transporters on those cells [86]. Excitatory amino acid transporter 2 (EAAT2), expressed by astrocytes, is the most important glutamate transporter and clears glutamate from the extracellular space. Homozygous mice deficient in glutamate transporter GLT-1 (the rodent ortholog of human EAAT2) express lethal spontaneous seizures and degeneration of hippocampal neurons [87]. Contrarily, increased expression of EAAT2 in the G93A SOD1 mouse has been reported to delay disease onset [88]. EAAT2 function in the cortex and spinal cord of familial and sporadic ALS patients has been reported to be reduced [89]. Aberrant RNA or oxidative stress is assumed to contribute to downregulation of EAAT2 at the transcriptional, translational or post-translational level [90,91].

Increased release of glutamate from presynaptic terminals has been speculated in ALS [92]. Overactivity of the presynaptic neuron can be induced through calcium dysregulation at terminals. The endoplasmic reticulum (ER) is an essential cellular compartment and stores Ca^2+^, and ER stress leads to the release of Ca^2+^ from the ER lumen [93]. Accumulation of TAR DNA-binding protein 43 (TDP-43) in the cytoplasm of motor neurons may potentially induce ER stress in ALS. As such, ER stress has been reported to play a central role in ALS pathogenesis [92]. Additionally, increased levels of cytosolic Ca^2+^ lead to the activation of ER stress [94]. Moreover, using in vitro models, the basal glutamate, released from spinal cord nerve terminals of mice expressing human SOD1 with the G93A mutation, is elevated compared to transgenic mice expressing the wild-type human SOD1 or to non-transgenic controls [95]. Combinations of these alterations potentially increase the release of glutamate, promoting a degenerative cascade.

A recent study investigated expression profiles of inhibitory receptors in the corticospinal motor neuron of hSOD1^G93A^ mice at the presymptomatic stage and disclosed altered expression of several types of subunits of GABA receptors [96]. These findings may also support evidence that shows hyperexcitability in ALS.

### 3.4. Fluid Biomarkers

The level of glutamate in fasting plasma has been reported to be increased in ALS, measured by high performance liquid chromatography (HPLC), especially in patients with spinal onset, longer disease duration and male gender [97]. Moreover, this finding was confirmed through advanced metabolomics techniques, showing significantly elevated glutamic acid in plasma [98]. Furthermore, the concentration of glutamate in cerebrospinal fluid was also increased [56]. Additionally, high glutamate concentrations in cerebrospinal fluid, as measured by HPLC, were associated with a spinal onset, impaired limb function and a higher rate of deterioration in muscle strength. Combined these findings suggest that elevations of CSF glutamate concentrations are related to a cellular insult within the spinal cord.

## 4. Increased Excitability of the Peripheral Nerve

### 4.1. Neurophysiological Evidence

Widespread fasciculations and muscle cramps are characteristic features experienced by patients with ALS [99]. Recent advances in muscle ultrasonography have identified that widespread and profuse fasciculations are specific to ALS [100,101,102,103,104,105]. This concept has been incorporated into the Awaji Criteria and more recently, the Gold Coast Criteria for ALS [106,107]. Needless to say, abundant electromyogram studies also showed widespread and profuse fasciculations in ALS [108]. Additionally, about 95% of ALS patients experience muscle cramps during the course of their disease [109,110,111], and it seems likely that a significant proportion of these symptoms are related to an increase in axonal excitability [100].

While routine nerve conduction studies measure conduction velocities and estimate the number of axons contributing to the amplitude of compound potentials, excitability measurements assess axonal membrane function and provide insight into underlying disease mechanisms [112]. Nerve excitability measurements in ALS patients have identified increased persistent sodium conductances and decreased potassium currents in motor nerves of ALS patients [113]. At a basic level, sodium currents are excitatory, while potassium currents are inhibitory, suggesting an increased excitability in the motor axon of ALS patients that likely contributes to symptoms of fasciculations and muscle cramps [114]. In terms of longitudinal approaches, this increased excitability becomes more prominent with disease evolution [115] and may contribute to the pattern of hand muscle atrophy observed in ALS patients. Specifically, the ALS split hand is characterized by prominent atrophy involving the thenar eminence, with a relatively preserved hypothenar region, which appears specific to ALS [116,117,118,119]. Excitability measurements have established that increased motor nerve excitability is more prominent in the median nerve compared with the ulnar nerve [120]. This difference in nerve excitability between the median and ulnar nerves has been speculated to contribute to this dissociated hand muscle atrophy. Finally, consistent with the motor predominance of ALS, assessment of peripheral sensory nerve excitability appears to be normal in ALS [121].

### 4.2. Pathological Evidence

A microarray analysis of spinal motor neurons in ALS patients determined a reduced expression of mRNA involving several types of potassium channels (KCNA1, KCNA2 and KCNQ2) [122]. Separate studies that investigated expressions of sodium and potassium channels in ventral and dorsal roots of ALS patients disclosed a markedly reduced expression of potassium channels in the ventral roots [123]. Consistent with functional neurophysiological approaches, these pathological findings are consistent with the development of motor hyperexcitability in ALS. Regarding potential causation, impaired transport of axonal membrane proteins, including proteins required for maintaining the integrity of potassium channels, seems likely to contribute to this altered expression of potassium channels.

## 5. Therapeutic Potential of Altered Neuronal Excitability in ALS

### 5.1. Ion Channel Modulators

Several studies have investigated the relationship between neuronal excitability and prognosis in ALS. Assessment of cortical function has identified that ALS patients with a hyperexcitable motor cortex, especially dysfunction of GABAergic neurons, have shorter survival compared to patients with preserved motor function [124]. Similarly, axonal excitability techniques have revealed that ALS patients with hyperexcitable peripheral nerves, especially increased persistent sodium currents and decreased potassium currents, have shorter survival with a more rapid functional decline [125,126]. In support, the neuroprotective agent riluzole partly suppresses cortical and axonal hyperexcitability (‘pseudonormalization’) in patients with ALS [127].

Based on these findings, suppression of cortical and axonal hyperexcitability in ALS has been attempted in several recent clinical trials, using ion channel modulators (Figure 2). Mexiletine is a class Ib antiarrhythmic drug and blocks sodium channels [128]. A clinical trial investigated the utility of mexiletine, with sixty patients randomly assigned (1:1) to riluzole 100 mg or riluzole plus mexiletine 300 mg [129]. The primary endpoint was set as changes in the revised ALS functional rating scale (ALSFRS-R) for six months, and the secondary endpoint was defined as strength–duration time constant (a measure of persistent sodium conductances), as determined by nerve excitability. This trial determined that 300 mg mexiletine failed to slow ALSFRS-R decline or suppress the strength–duration time constant. At the conclusion of this trial, it was speculated that higher doses of mexiletine would be required to suppress sodium currents. In terms of symptomatic approaches, a further clinical trial used mexiletine for muscle cramps in ALS [130]. Sixty patients with ALS were randomly allocated 1:1:1 to placebo, mexiletine 300 mg/day or mexiletine 900 mg/day, with longitudinal assessment for 12 weeks. This trial established significant reductions of muscle cramp frequency and cramp intensity in the mexiletine group and concluded that mexiletine treatment resulted in dose-dependent reductions in muscle cramp frequency and severity, with further support provided by a subsequent placebo-controlled crossover clinical trial [131]. Other sodium channel blocking agents have also been trialled, including flecanide, the latter by means of a double-blind, placebo-controlled, randomized clinical trial of 32 weeks duration, with a 12-week lead-in phase [132]. The slope of decline of ALSFRS-R during the treatment period was investigated as a primary endpoint. Additionally, nerve and cortical excitabilities were measured as a secondary endpoint. Fifty-four participants were randomly assigned to flecainide (26 patients) or placebo (28 patients) groups. No serious adverse events were reported in either group. There was no significant difference in the rate of decline in the primary endpoint, which measured ALSFRS-R between placebo and flecainide groups. Nerve excitability measurements revealed that changes in hyperpolarizing threshold electrotonus and current threshold relationship (markers of membrane potential) were milder in the flecainide group during the treatment period compared with those in the placebo group. This clinical trial concluded that flecainide may stabilize membrane potential. Additionally, riluzole has a few effects on several types of ion channels and modulates not only glutamate receptors but also sodium channels [133]. Blocking effects of the channel with riluzole should also be a focus.

Several clinical trials have focused on potassium channel modulators for ALS. As an exemplar, a recent double-blind, placebo-controlled randomized clinical trial examined the effect of ezogabine, a potassium channel opener, on cortical and peripheral excitability in a cohort of ALS patients [134]. A total of 65 ALS patients were allocated to placebo (23 participants), ezogabine 600 mg/day (23 participants) and ezogabine 900 mg/day (19 participants). Regarding changes in cortical function, ezogabine 900 mg/day increased SICI (ezogabine 600 mg/day did not change it), consistent with a decrease in cortical excitability. Regarding peripheral nerve function, ezogabine 900 mg/day suppressed a strength–duration time constant, and both ezogabine 600 mg/day and 900 mg/day increased rheobase. Additionally, ezogabine 900 mg/day increased depolarizing threshold electrotonus, and both ezogabine 600 mg/day and 900 mg/day decreased superexcitability. Taken in total, these alterations in nerve excitability are consistent with membrane hyperpolarization. In terms of safety profiles, drug tolerability rates were similar to those of ezogabine therapy for epilepsy, and further, more definitive studies are currently under consideration.

Several clinical trials have focused on the function of an AMPA receptor in ALS. For instance, a randomized, double-blind, placebo-controlled clinical trial analyzed the effect of perampanel (an AMPA receptor blocker) on cortical function in ALS [135]. A total of 10 patients received perampanel 4 mg/day, 5 received 8 mg/day and 4 received placebo. MT, as measured by single pulse TMS, was increased by perampanel administration, suggesting decreased cortical excitability. No side effects were reported. Two further clinical trials have investigated clinical utility of perampanel for ALS. One open label pilot study verified the safety and tolerability of perampanel for ALS [136]. A total of six patients with ALS was enrolled and received perampanel, which was gradually increased from 2 mg/day to 8 mg/day. All participants experienced adverse events, mostly behavioral events, and four participants withdrew from this trial due to adverse events. This study concluded that administration of perampanel for ALS was limited by its poor tolerability in high dose administration. Another randomized, double-blind, placebo-controlled, phase 2 clinical trial evaluated the efficacy and safety of perampanel for ALS [137]. Sixty-six patients were randomly allocated (1:1:1) to receive placebo, perampanel 4 mg/day or perampanel 8 mg/day for 48 weeks. A total of 18 participants for placebo, 14 participants for 4 mg perampanel and 7 participants for 8 mg perampanel completed the 48-week trial. Decline rates of ALSFRS-R in the perampanel 8 mg group were greater than those of placebo due to worsening of the bulbar subscore in the perampanel 8 mg group. Serious adverse events were more frequent in the perampanel 8 mg group than in the placebo group. This trial concluded perampanel was inefficacious to slow ALS progression.

Based on these clinical trials, it is concluded that several ion channel modulators may potentially modulate cortical and peripheral nerve hyperexcitability in ALS. What remains lacking to date are larger scale international studies with more regimented trial protocols and a focus on ALS disease progression [138,139]. It is further concluded that excitability techniques, both central and peripheral, may facilitate clinical trial design through their utility as functional biomarkers, potentially shortening trial durations and reducing the number of participants, ideally in a platform trial design.

### 5.2. Non-Pharmacological Approaches

Advanced neuromodulation techniques have also been used in a clinical trial setting to slow disease progression in ALS [140]. As an example, corticospinal excitability can be modulated through non-invasive brain stimulation (NIBS) techniques, such as transcranial direct current stimulation (tDCS) and repetitive TMS (rTMS) [141]. While the exact physiological mechanisms inducing alteration of corticospinal activity still remain unclear, most NIBS protocols selectively modulate I waves [37]. The interaction of the induced currents in the brain has an impact on interneuronal function connected to corticospinal neurons or via a reverberating local circuit within M1, composed of the superficial population of excitatory pyramidal neurons of layers II and III, the large pyramidal neurons in layer V and the inhibitory GABA cells.

In contrast, tDCS delivers constant and low (1–2 mA) electrical direct current via a circuit formed between two or more electrodes on the head and thereby modulates cortical excitability. Subthreshold membrane depolarization or hyperpolarization, as induced by this technique, can influence the neuron spontaneous firing, such as long-term potentiation (LTP) and long-term depression (LTD) synaptic plasticity [142]. Depending on protocols with different types of electrode size, placement, current and duration of stimulation, the overall effect may vary. Cathodal tDCS can suppress cortical excitability for several hours. As such, several clinical trials have attempted to slow ALS progression based around this technique. One clinical trial applied cathodal tDCS on M1 for two ALS patients to evaluate effects on disease progression [143]. Changes of ALSFRS-R in both patients were monitored before cathodal tDCS and during the period of cortical stimulation, and it determined that changes were similar before and during cathodal tDCS. Other clinical trials investigated the utility of tDCS based around the similar methodologies [144,145]. One trial included a single ALS patient, applied 12 sessions each of anodal, sham and cathodal tDCS for 4 weeks and measured changes of ALSFRS-R and mobility. No adverse events or change in disease progression were identified during the study period, but gait speed was slightly improved during anodal tDCS. Another trial incorporated long-term treatment using home-based protocols for remote supervision of anodal tDCS on M1. Two ALS patients were included and completed 24 remotely supervised anodal tDCS sessions for 8 weeks. No major adverse effects were revealed, but apparent clinical benefits were not discerned. From a physiological point of view, one study delivered three sessions of cathodal tDCS to M1, lasting 7, 11 or 15 min for 10 ALS patients and 10 healthy controls and investigated cortical excitability before and after the tDCS [146]. While cathodal tDCS induced a consistent decrease in cortical excitability in healthy controls, similar effects were not evident in ALS patients. Recently, a randomized, double-blind, sham controlled trial with tDCS for 30 ALS patients was performed [147]. Patients were randomly allocated to receive real or sham cortico-spinal tDCS (2:1) for 20 min/day and 5 days/week for 2 weeks. Two separate concurrent anodes were placed on the scalp over the motor cortex areas, and the cathode was placed over the spinal cervical enlargement (over C6). tDCS significantly improved muscle strength and scores for caregiver burden and quality of life and restored SICI and ICF, measured by paired pulse TMS. Additionally, these clinical improvements persisted at 6 months following the end of treatment and were associated with restoration of TMS indices. In conclusion, tDCS may modulate cortical excitability and mildly improve clinical findings, but the strength of those effects do not appear substantial.

A further technique, rTMS, delivers trains of pulses with a variety of frequencies, to induce prolonged effects on cortical excitability [148,149]. Specific patterns of magnetic stimulation may produce divergent responses. Basically, while low frequency constant-rate rTMS (<1 Hz) reduces cortical excitability, rTMS with higher frequency (>10 Hz) stimulation may increase cortical excitability. Additionally, a recently developed rTMS technique, termed theta burst stimulation (TBS), effectively alters cortical excitability over a short period. While continuous TBS (cTBS) generates a 40 s train of three pulses of 50 Hz stimulation repeated every 200 ms for a total of 600 stimuli and decreases cortical excitability, intermittent TBS (iTBS), with 10 bursts of high frequency stimulation, three pulses at 5 Hz, applied at 5 Hz every 10 s for a total of 600 pulses, increases cortical excitability. The utility of these techniques has been established in psychiatric disorders, especially depression, and they are widely used in clinical practice [150]. In ALS, several techniques have been assessed to slow ALS progression. One pilot clinical trial investigated the effectiveness of rTMS with low (1 Hz) and high (20 Hz) frequencies on M1 in ALS patients [151]. While low frequency rTMS was used to decrease cortical excitability, high frequency rTMS was tried to generate brain-derived neurotrophic factor (BDNF), because a prior study showed that high frequency (20 Hz) rTMS increases BDNF plasma levels in healthy subjects [152]. A total of four patients was included and allocated to low or high frequency rTMS (1:1) in this trial. Participants were evaluated before and during rTMS treatment, with Norris rating scale and muscle strength. rTMS was well tolerated, and patients with low frequency rTMS showed slightly slow deterioration in both evaluation items compared with before rTMS treatment. Moreover, an opposite tendency was observed in patients with high frequency rTMS, possibly mediated by increasing non-NMDA transmission [44]. Another pilot study utilized 5 Hz rTMS applied to the M1 region bilaterally over a 2-week period [153]. A total of 10 patients was assigned to active or sham stimulation (1:1), and they were evaluated with functional, fatigue, quality of life and muscle strength scales. At the end of active rTMS, quality of life and muscle strength were transiently improved, compared with sham groups, although these effects did not persist 2 weeks after discontinuation of rTMS. Similarly, cTBS was tested in several clinical trials. One double-blind, placebo-controlled trial enrolled 20 ALS patients and investigated the efficacy of cTBS. Participants were randomly allocated to active or sham stimulation (1:1) [154]. While five patients dropped out of this study, bilateral cTBS was delivered to M1 for 5 days/month for 6 months and slowed disease progression, measured by ALSFRS-R and muscle strength, in patients with active stimulation, compared to patients with sham stimulation. Based on this clinical trial, another trial was designed to disclose the usefulness of cTBS for ALS [155]. This trial was also performed as a double blind, placebo-controlled trial and delivered bilateral cTBS to M1 for 5 days/month for 1 year. A total of 20 ALS patients was enrolled and assigned to active or sham stimulation (1:1). However, this study failed to confirm the findings of the previous study and showed similar deterioration of ALSFRS-R between active and sham groups. The most prominent limitation to these clinical trials is the number of patients, because the numbers of participants were small.

While several clinical trials with NIBS for ALS patients may suggest potential clinical utility, there are several technical issues yet to be overcome. One of the main problems in the therapeutic use of NIBS relates to the duration of effectiveness. To drive a more persistent effectiveness of NIBS, repeated deliveries seem necessary. However, NIBS requires time, specific equipment and the assistance of trained staff. Separately, based on similar reasons, the number of participants in prior clinical trials has remained relatively small. As such, it seems likely that simpler and clinically suitable equipment will need to be developed. Additionally, a remotely supervised NIBS system will be required. After solving these more technical issues, NIBS potentially combined with pharmaceutical interventions, including ion channel modulators, may potentially slow ALS progression, akin to the utility of these techniques in psychiatric disorders, particularly depression.

## Figures and Tables

**Figure 1 pharmaceuticals-15-00433-f001:**
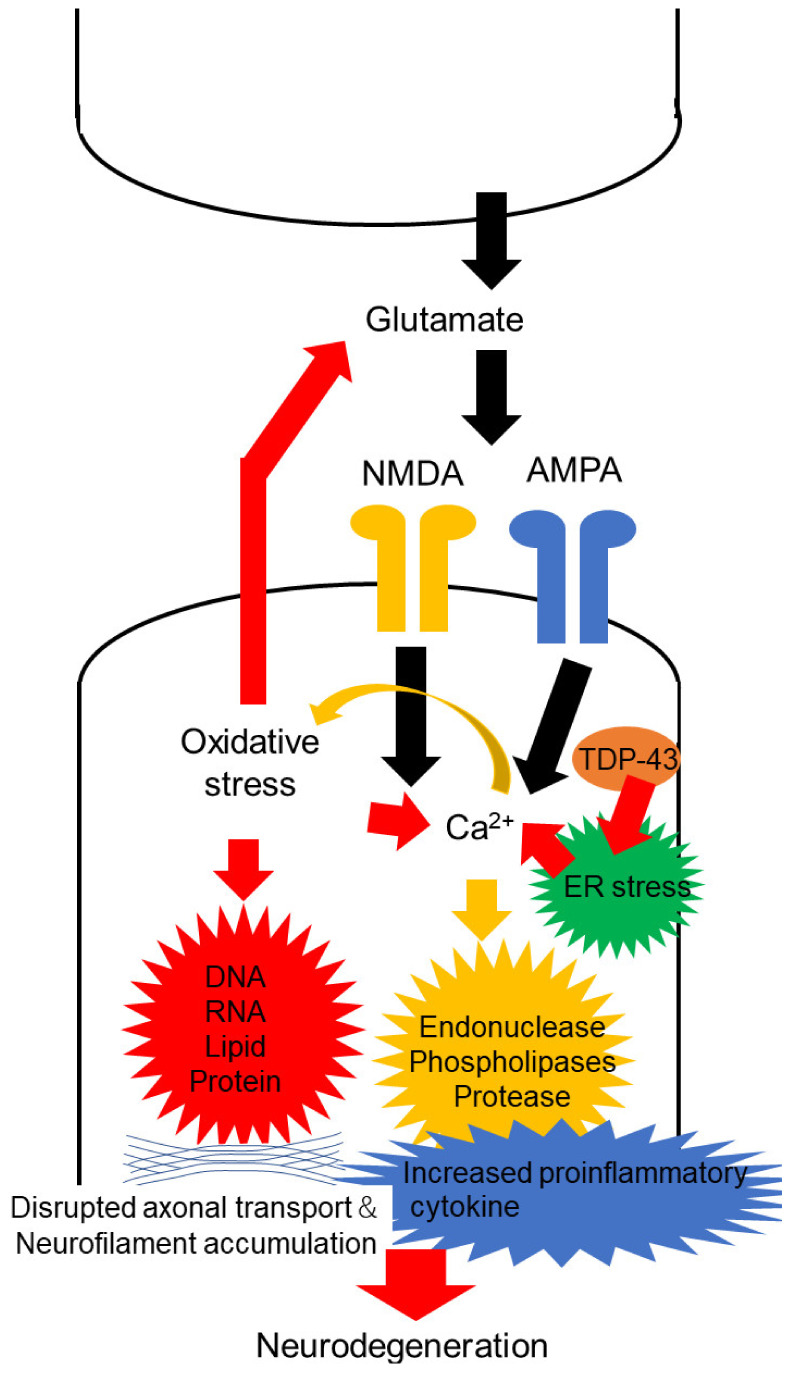
Neuronal excitability and free radicals accelerate motor neuron degeneration in ALS. Excessive activation of α-amino-3-hydroxy-5-methyl-4-isoxazolepropionate (AMPA) and N-methyl-D-aspartate (NMDA) receptors, mediated by glutamate, increase the influx of calcium ions (Ca^2+^). Ca^2+^ activates enzymes such as endonucleases, phospholipases and proteases, which may induce neuronal injury. Additionally, excessive activation of glutamate receptors generates free radicals and oxidative stress to thereby modulate DNA, RNA, lipids and proteins, increase glutamate concentration in the synaptic cleft and raise Ca^2+^ influx into the cells. These processes accumulate neurofilaments and increase proinflammatory cytokines, which might result in neurodegeneration. Moreover, accumulation of TAR DNA-binding protein 43 (TDP-43) in the cytoplasm of motor neurons potentially induces ER stress, and ER stress leads to the release of Ca^2+^ from the ER lumen.

**Figure 2 pharmaceuticals-15-00433-f002:**
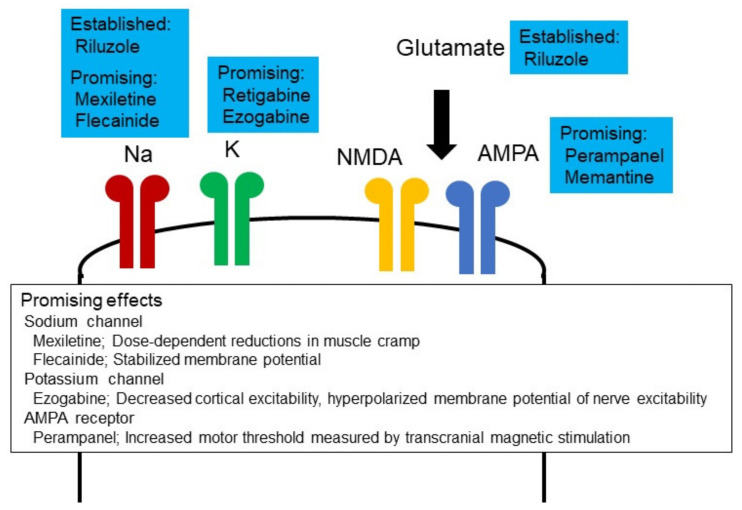
Neuronal excitability and therapeutic development. Ion channel modulators have been investigated for efficacy in ALS. Riluzole inhibits glutamate release, enhances glutamate uptake, blocks voltage-dependent sodium channels, antagonizes NMDA receptors and inhibits GABA uptake. Perampanel exerts effects via AMPA receptors, retigabine and ezogabine through potassium channels and mexiletine and flecainide via sodium channels (see text for clinical trial results). These ion channel modulators showed several lines of evidence.

## Data Availability

Not applicable.

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
