# Peer review of "Neuronal Hyperexcitability and Free Radical Toxicity in Amyotrophic Lateral Sclerosis: Established and Future Targets"

_pharmaceuticals, 2022, doi:10.3390/ph15040433_

Round 1

Reviewer 1 Report

In the current manuscript, the authors have carefully described available literature on the relationship between glutamate and ALS. This review is very timely and has lots of good information covered, including the recent therapeutic developments. Overall, the structure of the paper looks good, but I have some suggestions, which authors should consider while revising the paper.

References are missing for some important statements:

Page 2, last para, ‘This influx of….cell death’ needs citation.

Page 3, last para, ‘Moreover, ROS……neurodegenerative processes’ needs citation.

There are various other places, where authors should cite some original papers supporting the statements.

The paper would benefit if authors can make a table to summarize various evidences described in the text. In table format, it would be easier to follow all the information. Another table for the mentioned drugs (in figure 2) and other therapeutic options, clinical trials and results may help better presentation of the information to the readers.

Figures are very basic and do not contribute much to the manuscript. I would recommend adding mechanistic pathways to the figures to show how:

Figure 1: glutamate excitotoxicity affects various cellular pathways contributing to the pathological changes.

Figure 2. targeting specific receptors contribute to therapeutic benefits shown by the shown drugs.

 Please incorporate these changes, which I believe could add value to the manuscript.

Author Response

Reviewer: 1

Comments and Suggestions for Authors

In the current manuscript, the authors have carefully described available literature on the relationship between glutamate and ALS. This review is very timely and has lots of good information covered, including the recent therapeutic developments. Overall, the structure of the paper looks good, but I have some suggestions, which authors should consider while revising the paper.

References are missing for some important statements:

Page 2, last para, ‘This influx of….cell death’ needs citation.

Response: The following citation has been added.

Lau, A.; Tymianski, M. Glutamate receptors, neurotoxicity and neurodegeneration. Pflugers Archiv : European journal of physiology 2010, 460, 525-542.

Page 3, last para, ‘Moreover, ROS……neurodegenerative processes’ needs citation.

Response: The following citation has been added.

Rao, S.D.; Weiss, J.H. Excitotoxic and oxidative cross-talk between motor neurons and glia in als pathogenesis. Trends in neurosciences 2004, 27, 17-23.

There are various other places, where authors should cite some original papers supporting the statements.

Response: We appreciate the reviewer’s comments. We have tried to put appropriate citation.

The paper would benefit if authors can make a table to summarize various evidences described in the text. In table format, it would be easier to follow all the information. Another table for the mentioned drugs (in figure 2) and other therapeutic options, clinical trials and results may help better presentation of the information to the readers.

Response: We thank for the reviewer’s comments. The following tables, which have described several pieces of evidence, have been added.

Figures are very basic and do not contribute much to the manuscript. I would recommend adding mechanistic pathways to the figures to show how:

Figure 1: glutamate excitotoxicity affects various cellular pathways contributing to the pathological changes.

Figure 2. targeting specific receptors contribute to therapeutic benefits shown by the shown drugs.

Response: We thank for the reviewer’s helpful comments. According to reviewer’s advice, figures have been revised.

Please incorporate these changes, which I believe could add value to the manuscript.

Reviewer 2 Report

his review focuses on the relationship between free radicals and excitotoxicity in motor neuronal degeneration in ALS. However, there are still some questions, which should be revised.

1. In the manuscript, authors mention that glutamate is the excitatory neurotransmitter, that binds to several AMPA and NMDA receptors. Excessive activation of these receptors generates FRs, inducing neurodegeneration. Then what is the relationship between FRs and excitotoxicity in motor neuronal degeneration in ALS? Excessive activation of these receptors are the upstream of FRs? Or there are different mechanisms underlying the FRs and excitotoxicity in motor neuronal degeneration in ALS?

2. The subtitle of “Motor neuron degeneration in ALS” is not consistent with the content. In this part, authors mainly demonstrated the role of ROS in Motor neuron degeneration in ALS.

3. The subtitle of “Increased neuronal excitability and cortical function” is inappropriate. Authors demonstrated mainly the cortical hyperexcitability in ALS, not increased cortical function. And the author describes more on the evidence about increased neuronal excitability in ALS. However, the title is

“Glutamate and free radical toxicity in amyotrophic lateral sclerosis: established and future targets”, which seems to demonstrate the possible mechanisms underlying the Glutamate and free radical toxicity in ALS.

The authors needs to revise the manuscript and make it more clearly.

4. There are many sentences, which have no references in the manuscript. For example, “Moreover, ROS increases glutamate concentrations, raises Ca2+ influx into the cells and mitochondria and leads to impair glial function. Additionally, ROS may activate glia and induce proinflammatory cytokine production. ”

Author Response

Reviewer: 2

Comments and Suggestions for Authors

his review focuses on the relationship between free radicals and excitotoxicity in motor neuronal degeneration in ALS. However, there are still some questions, which should be revised.

  1. In the manuscript, authors mention that glutamate is the excitatory neurotransmitter, that binds to several AMPA and NMDA receptors. Excessive activation of these receptors generates FRs, inducing neurodegeneration. Then what is the relationship between FRs and excitotoxicity in motor neuronal degeneration in ALS? Excessive activation of these receptors are the upstream of FRs? Or there are different mechanisms underlying the FRs and excitotoxicity in motor neuronal degeneration in ALS?

Response: We thank for the reviewer’s helpful comments. ROS, induced by excessive activation of glutamate receptors, increases glutamate concentrations, raises Ca2+ influx into the cells and mitochondria. As such, these mechanisms are tightly related. These explanations have been clearly described in the revised manuscript.

“Moreover, ROS disrupts glutamate transport in surrounding astrocytes, increases glutamate concentrations in the synaptic cleft, raises Ca2+ influx into the cells and mitochondria and leads to impair glial function [27].”

“In conclusion, excessive activation of glutamate receptors induces oxidative stress, and oxidative stress can cause neurodegeneration via various mechanisms. Moreover, oxidative stress increases glutamate concentrations and raises Ca2+ influx. As such, the excitatory neurotransmitter and oxidative stress are tightly connected with each other and might be substantially related with neurodegeneration in ALS.”

  1. The subtitle of “Motor neuron degeneration in ALS” is not consistent with the content. In this part, authors mainly demonstrated the role of ROS in Motor neuron degeneration in ALS.

Response: Revised as suggested. “Motor neuron degeneration in ALS” has been revised to “Oxidative stress in motor neuron degeneration of ALS”.

  1. The subtitle of “Increased neuronal excitability and cortical function” is inappropriate. Authors demonstrated mainly the cortical hyperexcitability in ALS, not increased cortical function. And the author describes more on the evidence about increased neuronal excitability in ALS. However, the title is

“Glutamate and free radical toxicity in amyotrophic lateral sclerosis: established and future targets”, which seems to demonstrate the possible mechanisms underlying the Glutamate and free radical toxicity in ALS.

The authors needs to revise the manuscript and make it more clearly.

Response: We thank for the reviewer’s helpful comments. We have added several sentences, including the following sentence, which have explained the relationship between glutamate excitotoxicity and free radicals. Additionally, the title and subtitles have been revised.

“In conclusion, excessive activation of glutamate receptors induces oxidative stress, and oxidative stress can cause neurodegeneration via various mechanisms. Moreover, oxidative stress increases glutamate concentrations and raises Ca2+ influx. As such, the excitatory neurotransmitter and oxidative stress are tightly connected with each other and might be substantially related with neurodegeneration in ALS.”

Title; “Neuronal hyperexcitability and free radical toxicity in amyotrophic lateral sclerosis: established and future targets”

Subtitle; “Increased cortical excitability”

  1. There are many sentences, which have no references in the manuscript. For example, “Moreover, ROS in[1]creases glutamate concentrations, raises Ca2+ influx into the cells and mitochondria and leads to impair glial function. Additionally, ROS may activate glia and induce proinflammatory cytokine production. ”

Response: We thank for the reviewer’s helpful comments. Several references, including the following citation, have been added in the revised manuscript.

Rao, S.D.; Weiss, J.H. Excitotoxic and oxidative cross-talk between motor neurons and glia in als pathogenesis. Trends in neurosciences 2004, 27, 17-23.

Reviewer 3 Report

This review paper deals with the sources of excitotoxicity induced by glutamate and free radicals in ALS and related therapeutic targets.

Main comments:

p2, last pgph, lines 3-6 – the cited reference Bondy & Lee (as old as from 1993) actually deals with free radicals generated by glutamate agonists. There is no explanation in that paper nor by the authors on the mechanism or at least for the ocurence of „increased post-synaptic sensitivity to glutamate“.

p3 line 7 from bottom . what are „glial synapses“? In the next line authors write „ROS increases glutamate concentration“. How does this come about. Is the release of glutamate augmented or are the uptake mechanisms affected. This should be clarified.

p4 line 7 – what is the “triple structure”?

p4 line 3 from bottom – “….increased excitability via alteration of these neurotransmitters…” Which are “these neurotransmitters”? If this refers to the 3 transmitters mentioned in the previous sentence (Glu, Nora, GABA) it should be explained how the alteration of these very different synapses acts to cause increased MEPs. There are also no references cited for this statement.

p5&6 – chapter: Pathological and animal model evidence – The references cited are rather old. New studies should be referred (eg. Jara, J. H., Sheets, P. L., Nigro, M. J., Perić, M., Brooks, C., Heller, D. B., Martina, M., Andjus, P. R., & Ozdinler, P. H. (2020). The Electrophysiological Determinants of Corticospinal Motor Neuron Vulnerability in ALS. Frontiers in molecular neuroscience, 13, 73. https://doi.org/10.3389/fnmol.2020.00073)

p.6 , 2nd pgph line 8-9, what is “basal glutamate efflux” from nerve terminals? Did the authors mean spontaneous release of glutamate? Release is not efflux.

p6, chapter Neurophysiological evidence – there is no mention of EMG which is the classical clinical e-physiological diagnostic technique.

p7, 3rd pgph, line 8 - riluzole can not be considered in a chapter on Ion channel modulators. Its mechanism of action is complex, and can only vaguely be considered to block Na channels (however this is not even mentioned in the cited paper)

p7, last pgph, line 3 - Mexiletine is mentioned but no explanation of its mode of action

Fig. 2 –its graphics is not informative. A Table also baring relevant references would be more useful.

Minor comments:

p2 line 8 – it should b SOD1 that is aggregated (not SOD)

Legend to Fig. 1 – Ca2+ (instead of Ca2+); line 6 should read „increase glutamate concentration in the synaptic cleft“

p4 line 3-5 – This sentence is hardly legible.

p7, last pgph, line 2 – “channel” instead of “cannel”

p8, last pgph, line 7-8 – something is inconsistent in this sentence – “ezogabine 600 mg/day did not change it” should maybe be put in brackets.

Author Response

Reviewer: 3

Comments and Suggestions for Authors

This review paper deals with the sources of excitotoxicity induced by glutamate and free radicals in ALS and related therapeutic targets.

Main comments:

p2, last pgph, lines 3-6 – the cited reference Bondy & Lee (as old as from 1993) actually deals with free radicals generated by glutamate agonists. There is no explanation in that paper nor by the authors on the mechanism or at least for the ocurence of „increased post-synaptic sensitivity to glutamate“.

Response: Citation has been revised. The following citation has been added.

Lau, A.; Tymianski, M. Glutamate receptors, neurotoxicity and neurodegeneration. Pflugers Archiv : European journal of physiology 2010, 460, 525-542.

p3 line 7 from bottom . what are „glial synapses“? In the next line authors write „ROS increases glutamate concentration“. How does this come about. Is the release of glutamate augmented or are the uptake mechanisms affected. This should be clarified.

Response: We thank for the reviewer’s helpful comments. “glial synapses“ has been revised to “glial cell synapses”. ROS has been reported to disrupt glutamate transport in surrounding astrocytes and result in increasing glutamate in the synaptic cleft. This explanation has been added in the revised manuscript.

“Moreover, ROS disrupts glutamate transport in surrounding astrocytes, increases glutamate concentrations in the synaptic cleft, raises Ca2+ influx into the cells and mitochondria and leads to impair glial function [27].”

Rao SD, Weiss JH. Excitotoxic and oxidative cross-talk between motor neurons and glia in ALS pathogenesis. Trends Neurosci. 2004 Jan;27(1):17-23.

p4 line 7 – what is the “triple structure”?

Response: “triple structure” has been revised to “the neurofilament triple proteins”.

p4 line 3 from bottom – “….increased excitability via alteration of these neurotransmitters…” Which are “these neurotransmitters”? If this refers to the 3 transmitters mentioned in the previous sentence (Glu, Nora, GABA) it should be explained how the alteration of these very different synapses acts to cause increased MEPs. There are also no references cited for this statement.

Response: We thank for the reviewer’s helpful comments. Several lines of evidence have reported increased glutamatergic and noradrenergic and decreased GABAergic neurotransmitters in ALS patients. These alterations, suggesting increased excitability, might contribute to increased MEP amplitudes. This explanation with citation has been added in the revised manuscript.

“Supplementally, several lines of evidence have disclosed increased glutamatergic and noradrenergic and decreased GABAergic neurotransmitters in ALS patients [54-56]. These alterations, suggesting increased excitability, might contribute to increased MEP amplitudes.”

Bertel, O.; Malessa, S.; Sluga, E.; Hornykiewicz, O. Amyotrophic lateral sclerosis: Changes of noradrenergic and serotonergic transmitter systems in the spinal cord. Brain research 1991, 566, 54-60.

Chew, S.; Atassi, N. Positron emission tomography molecular imaging biomarkers for amyotrophic lateral sclerosis. Frontiers in neurology 2019, 10, 135.

Spreux-Varoquaux, O.; Bensimon, G.; Lacomblez, L.; Salachas, F.; Pradat, P.F.; Le Forestier, N.; Marouan, A.; Dib, M.; Meininger, V. Glutamate levels in cerebrospinal fluid in amyotrophic lateral sclerosis: A reappraisal using a new hplc method with coulometric detection in a large cohort of patients. Journal of the neurological sciences 2002, 193, 73-78.

p5&6 – chapter: Pathological and animal model evidence – The references cited are rather old. New studies should be referred (eg. Jara, J. H., Sheets, P. L., Nigro, M. J., Perić, M., Brooks, C., Heller, D. B., Martina, M., Andjus, P. R., & Ozdinler, P. H. (2020). The Electrophysiological Determinants of Corticospinal Motor Neuron Vulnerability in ALS. Frontiers in molecular neuroscience, 13, 73. https://doi.org/10.3389/fnmol.2020.00073)

Response: We thank for the reviewer’s helpful comments. This citation has been added, with some comments.

“A recent study investigated expression profiles of inhibitory receptors in corticospinal motor neuron of hSOD1G93A mice at the presymptomatic stage and disclosed altered expression of several types of subunits of GABA receptors [96]. These findings may also support evidence which show hyperexcitability in ALS.”

p.6 , 2nd pgph line 8-9, what is “basal glutamate efflux” from nerve terminals? Did the authors mean spontaneous release of glutamate? Release is not efflux.

Response: Revised as suggested. ”Release” is correct.

p6, chapter Neurophysiological evidence – there is no mention of EMG which is the classical clinical e-physiological diagnostic technique.

Response: We thank for the reviewer’s helpful comments. Comments on EMG have been added in the revised manuscript.

“Needless to say, abundant electromyogram studies also showed widespread and profuse fasciculations in ALS [108].”

p7, 3rd pgph, line 8 - riluzole can not be considered in a chapter on Ion channel modulators. Its mechanism of action is complex, and can only vaguely be considered to block Na channels (however this is not even mentioned in the cited paper)

Response: We thank for the reviewer’s useful comments. Effects on sodium channel with riluzole have been added in this paragraph with appropriate citation.

“Additionally, riluzole has a few effects on several types of ion channels and modulates not only glutamate receptors but also sodium channels [134]. Blocking effects of channel with riluzole should also be focused.”

p7, last pgph, line 3 - Mexiletine is mentioned but no explanation of its mode of action

Response: Added as suggested.

“Based on these findings, suppression of cortical and axonal hyperexcitability in ALS has been attempted in several recent clinical trials, using ion channel modulators (Figure2). Mexiletine a class Ib antiarrhythmic drug and blocks sodium channels [129].”

Fig. 2 –its graphics is not informative. A Table also baring relevant references would be more useful.

Response: According to comments from the reviewer #1, figures have been revised, and tables have been added.

Minor comments:

p2 line 8 – it should b SOD1 that is aggregated (not SOD)

Response: Revised as suggested.

Legend to Fig. 1 – Ca2+ (instead of Ca2+); line 6 should read „increase glutamate concentration in the synaptic cleft“

Response: Revised as suggested.

p4 line 3-5 – This sentence is hardly legible.

Response: Revised as follows.

“In turn, ROS released from damaged neurons influence surrounding glial cell synapses [27]. Moreover, ROS disrupts glutamate transport in surrounding astrocytes, increases glutamate concentrations in the synaptic cleft, raises Ca2+ influx into the cells and mitochondria and leads to impair glial function [27].”

p7, last pgph, line 2 – “channel” instead of “cannel”

Response: Revised as suggested.

p8, last pgph, line 7-8 – something is inconsistent in this sentence – “ezogabine 600 mg/day did not change it” should maybe be put in brackets.

Response: This part has been put in brackets.

“Regarding changes in cortical function, ezogabine 900 mg/day increased SICI (ezogabine 600 mg/day did not change it), consistent with a decrease in cortical excitability.”

Round 2

Reviewer 2 Report

No comments.

Author Response

Reviewer: 2

Comments and Suggestions for Authors

No comments.

Response: We thank for the reviewer.

Reviewer 3 Report

My review report is attached as a Word file.

I could not however verify the new tables and figures for the ms. 

----------------

The paper is well improved however a few points still need to be readdressed (see comments in red):

p2, last pgph, lines 3-6 – the cited reference Bondy & Lee (as old as from 1993) actually deals with free radicals generated by glutamate agonists. There is no explanation in that paper nor by the authors on the mechanism or at least for the ocurence of „increased post-synaptic sensitivity to glutamate“.

Response: Citation has been revised. The following citation has been added.

Lau, A.; Tymianski, M. Glutamate receptors, neurotoxicity and neurodegeneration. Pflugers Archiv : European journal of physiology 2010, 460, 525-542.

* Still I do not see any explanation for the statement „increased post-synaptic sensitivity to glutamate.“ Rather this should be related to glutamate receptor overstimulation.

p3 line 7 from bottom . what are „glial synapses“? In the next line authors write „ROS increases glutamate concentration“. How does this come about. Is the release of glutamate augmented or are the uptake mechanisms affected. This should be clarified.

Response: We thank for the reviewer’s helpful comments. “glial synapses“ has been revised to “glial cell synapses”.

* This surely is not offering an explanation why should glia have synapses that are neuronal characteristics.

p4 line 7 – what is the “triple structure”?

Response: “triple structure” has been revised to “the neurofilament triple proteins”.

The expression is “triplet”, however modern molecular biology states there are in fact  two more proteins in the neurofilaments.

p7, last pgph, line 3 - Mexiletine is mentioned but no explanation of its mode of action

Response: Added as suggested.

“Based on these findings, suppression of cortical and axonal hyperexcitability in ALS has been attempted in several recent clinical trials, using ion channel modulators (Figure2). Mexiletine a class Ib antiarrhythmic drug and blocks sodium channels [129].”

* OK but last line should read “Mexiletine is a class …”

Author Response

Reviewer: 3

Comments and Suggestions for Authors

The paper is well improved however a few points still need to be readdressed (see comments in red):

p2, last pgph, lines 3-6 – the cited reference Bondy & Lee (as old as from 1993) actually deals with free radicals generated by glutamate agonists. There is no explanation in that paper nor by the authors on the mechanism or at least for the ocurence of „increased post-synaptic sensitivity to glutamate“.

Response: Citation has been revised. The following citation has been added.

Lau, A.; Tymianski, M. Glutamate receptors, neurotoxicity and neurodegeneration. Pflugers Archiv : European journal of physiology 2010, 460, 525-542.

* Still I do not see any explanation for the statement „increased post-synaptic sensitivity to glutamate.“ Rather this should be related to glutamate receptor overstimulation.

Response: We thank for the reviewer’s comments. This description had mentioned “GluR2 mRNA editing”. A prior study disclosed incomplete GluR2 mRNA editing in ALS tissue, potentially resulting in increased permeability of Ca2+. However, as the reviewer described, “increased post-synaptic sensitivity“ might not be appropriate. As such, this sentence has been revised as follows.

“Excessive activation of these receptors, with defects in the clearance of these neurotransmitters from the synaptic cleft, combined with increased permeability of Ca2+, accumulates excitatory mediators, generates FRs and impairs neurons”

Kawahara Y, Ito K, Sun H, Aizawa H, Kanazawa I, Kwak S. Glutamate receptors: RNA editing and death of motor neurons. Nature. 2004 Feb 26;427(6977):801.

p3 line 7 from bottom . what are „glial synapses“? In the next line authors write „ROS increases glutamate concentration“. How does this come about. Is the release of glutamate augmented or are the uptake mechanisms affected. This should be clarified.

Response: We thank for the reviewer’s helpful comments. “glial synapses“ has been revised to “glial cell synapses”.

* This surely is not offering an explanation why should glia have synapses that are neuronal characteristics.

Response: Sorry for our inappropriate description. “glial cell synapses” has been revised to “glial cells”.

“In turn, ROS released from damaged neurons influence surrounding glial cells.”

p4 line 7 – what is the “triple structure”?

Response: “triple structure” has been revised to “the neurofilament triple proteins”.

The expression is “triplet”, however modern molecular biology states there are in fact  two more proteins in the neurofilaments.

Response: We thank for the reviewer’s helpful comments. The sentence has been revised as follows.

“Linked to this accumulation, ROS produced by mice with SOD1-mutations, have been reported mainly to bind to neurofilaments, disrupting the neurofilament proteins, while promoting neurofilament aggregation.”

p7, last pgph, line 3 - Mexiletine is mentioned but no explanation of its mode of action

Response: Added as suggested.

“Based on these findings, suppression of cortical and axonal hyperexcitability in ALS has been attempted in several recent clinical trials, using ion channel modulators (Figure2). Mexiletine a class Ib antiarrhythmic drug and blocks sodium channels [129].”

* OK but last line should read “Mexiletine is a class …”

Response: Revised as suggested.
